# Ammonium Fluoride as Suitable Additive for HILIC-Based LC-HRMS Metabolomics

**DOI:** 10.3390/metabo9120292

**Published:** 2019-11-27

**Authors:** Luca Narduzzi, Anne-Lise Royer, Emmanuelle Bichon, Yann Guitton, Corinne Buisson, Bruno Le Bizec, Gaud Dervilly-Pinel

**Affiliations:** 1Laboratoire d’Etude des Résidus et Contaminants dans les Aliments (LABERCA), Oniris, INRA, F-44307 Nantes, France; anne-lise.royer@oniris-nantes.fr (A.-L.R.); emmanuelle.bichon@oniris-nantes.fr (E.B.); yann.guitton@oniris-nantes.fr (Y.G.); bruno.lebizec@oniris-nantes.fr (B.L.B.); gaud.dervilly@oniris-nantes.fr (G.D.-P.); 2Département des analyses, Agence Française de Lutte contre le Dopage (AFLD), 92290 Châtenay-Malabry, France; c.buisson@afld.fr

**Keywords:** HILIC, metabolomics, mobile phase modifier, ammonium fluoride

## Abstract

Hydrophilic Interaction Liquid Chromatography (HILIC) chromatography is widely applied in metabolomics as a complementary strategy to reverse phase chromatography. Nevertheless, it still faces several issues in terms of peak shape and compounds ionization, limiting the automatic de-convolution and data semi-quantification performed through dedicated software. A way to improve the chromatographic and ionization performance of a HILIC method is to modify the electrostatic interactions of the analytes with both mobile and stationary phases. In this study, using a ZIC-HILIC chromatographic phase, we evaluated the performance of ammonium fluoride (AF) as additive salt, comparing its performance to ammonium acetate (AA). Three comparative criteria were selected: (1) identification and peak quality of 34 standards following a metabolomics-specific evaluation approach, (2) an intraday repeatability test with real samples and (3) performing two real metabolomics fingerprints with the AF method to evaluate its inter-day repeatability. The AF method showed not only higher ionization efficiency and signal-to-noise ratio but also better repeatability and robustness than the AA approach. A tips and tricks section is then added, aiming at improving method replicability for further users. In conclusion, ammonium fluoride as additive salt presents several advantages and might be considered as a step forward in the application of robust HILIC methods in metabolomics.

## 1. Introduction

Hydrophilic Interaction Liquid Chromatography (HILIC [1]) has gained popularity in LC-HRMS metabolomics due to its complementarity with Reverse Phase (RP) approaches [2]. Thanks to the water-mediated separation of the analytes, the HILIC technology allows to extend the separative abilities of liquid chromatography to very polar metabolites, increasing the metabolome coverage, while maintaining the robustness and the repeatability (critical parameters in metabolomics) at similar level than the RP chromatography [3].

In the metabolomics field, HILIC chromatographic separation is commonly achieved through an organic to aqueous gradient, consisting of acetonitrile (ACN) and ultra-pure water (H_2_O), assisted with small volatiles salts as additives, like ammonium acetate or ammonium formate [4], to improve chromatographic peak shape. To take advantage of the buffering/protonation effect of these salts, the corresponding acidic counterpart might be added in small amounts to achieve the respective pKa (pH 4.7 for ammonium acetate and pH 3.2 for ammonium formate, [5]). These methodological adjustments allow to achieve efficient chromatographic separation for the majority of the eluting peaks.

Ammonium formate and ammonium acetate have limited solubility in organic solvents [6,7]. Furthermore, they show poor analytes’ peak shape and ion suppression for some compounds [8,9]. The poor peak shape and the ion suppression of the compounds undermine the automatic peak detection performed in the metabolomics studies [10]. Recently, addition of micro molar concentrations of ammonium phosphate in HILIC has been demonstrated to have a great effect on the chromatographic peak shapes, improving also (to some extent) MS signal intensity [11]. On the other hand, some post-column modifiers have been also proposed to improve compound ionization, like 2 propionic acid-containing iso-propanol for ESI+ [12] and 2-(2-methoxyethoxy)-ethanol for ESI– [9,13]. Nonetheless, the latter are incompatible with the modern, polarity-switching, MS-methods.

In RP, to limit ion suppression due to the water-rich starting phase, different additives having ionization enhancing effects have been proposed. Among those, ammonium fluoride is currently gaining popularity in RP metabolomics due to its ability to improve ionization on average between 4–11 folds more than ammonium formate or formic acid, with singular compounds even showing increased ionization up to 22 folds, consequently increasing the number of compounds detected [14]. The enhanced ionization effect is due to the highest electronegativity of the fluoride ion. While ammonium fluoride has been originally used to improve negative ionization [14], better ionization properties have been also observed in the positive mode [15,16], due to the reverse ionization effect [17].

Apart from reversed phase chromatography, ammonium fluoride has already been tested as chromatographic modifier in Aqueous Normal Phase chromatography (ANP [18]) showing better ionization properties than ammonium formate for a range of compounds (NAD, trans-acotinic acid, 3-hydroxy glutaric acid, 3-methyl adipic acid, L-threonine and *N*-acetyl-carnitine); given the similarity between ANP and HILIC chromatographic outcome [19], it is interesting to test whether ammonium fluoride could also be a suitable additive in HILIC-based metabolomics experiments. The suggested concentration of ammonium fluoride in the mobile phase is between 1 to 2 mM [14,18]; this value is more compatible with high organic solvent used in HILIC in comparison to the standard 10 mM concentration used for ammonium acetate [7].

In this study, we tested a milli-molar concentration of ammonium fluoride as an alternative modifier in HILIC, comparing the performances to a robust standard ammonium acetate HILIC method [20], to discuss whether ammonium fluoride is relevant in HILIC based metabolomics. We evaluated its performance in terms of peak shape, peak intensity and peak area, but also repeatability and robustness, injecting also several samples to evaluate the intra-day variability. Then, as proof of concept, a set of human bio-fluids collected within a previous framework have been characterized using a metabolomics experimental design, similar to the work already carried on RP for the same two sample-sets (Narduzzi et al. 2019). Finally, a tips and tricks section is added, to help end-users to avoid common mistakes when working with a “relative unknown” modifier like ammonium fluoride.

## 2. Materials and Methods

### 2.1. Solvents and Reagents

All solvents and reagents used in this study were of analytical quality. Acetonitrile (ACN), methanol (MeOH), acetic acid (C_2_H_4_O_2_^−^) and ammonia (NH_4_) were purchased from Honeywell (Bucharest, Romania). Ultra-pure water (H_2_O) was purchased from VWR (Fontenay-sous-Bois, France); Chloroform (CHCl_3_) was purchased from Carlo Erba Reactifs (SDS, Peypin, France). Ammonium Fluoride (AF) was purchased from Acros Organics (Geel, Belgium). Metabolomics isotope labeled internal standards (l-Leucine-5,5,5-d3, l-Tryptophan-2,3,3-d3, Indole-2,4,5,6,7-d5-3-acetic acid and 1,14-Tetradecanedioic-d24 acid) were from Sigma–Aldrich (Saint Quentin Fallavier, France) and CDN Isotopes (Pointe-Claire, QC, Canada). MSCAL6 ProteoMass LTQ/FT-Hybrid, standard mixtures used for calibration of the MS instrument, were obtained from Sigma–Aldrich (Saint Quentin Fallavier, France).

### 2.2. Standards Preparation

Stock solutions of each standard (1 mg mL^−1^) were prepared in a suitable solvent mixture to obtain an optimal solubility. When necessary, stock solutions were slightly acidified or alkalinized to allow the compound to solubilize in the mixture. Working solutions at 5 ng mL-1 were then prepared by diluting the stock solutions in a mixture of water containing 0.1% acetic acid and acetonitrile (1/1). All the standards were purchased from Sigma–Aldrich (Saint Quentin Fallavier, France).

### 2.3. Sample Preparation

The sample-sets and the experimental design applied in this study have been described in Narduzzi et al. 2019, and consisted of the urine and plasma of 16 volunteer men treated either with micro-doses of Erythropoietin (Treat 1) or micro-doses of EPO + growth hormone (Treat 2). The preparation of both matrices has been minimal, similarly to any metabolomics experiment, to keep the highest number of metabolites while reducing the variability induced by the sample treatment.

*Urine:* The urine samples gravity was determine using a digital urine specific gravity refractometer (4410 PAL-10S, Cole-armer, Kingwood, TX, USA). The samples were normalized with ultra-pure water and further filtered through 10 kDa filters (VWR centrifugal tubes, modified polyethersulfone 10 kDa, 500 lL, VWR, Tultitlán de Mariano Escobedo, Mexico) polyether-sulfone membranes under centrifugation at 10,000× *g* at 5 °C for 20 min [21]. Deuterated internal standards consisting of L-leucine-5,5,5-d3, L-tryptophan-2,3,3-d3, Indole-2,4,5,6,7-d5-3-acetic acid, 1,14-tetradecanedioic-d24-acid were diluted to 5 ppm and added in each sample after the extraction.

*Plasma:* Plasma samples were extracted using the protocol of Jacob et al. [21]; a tri-phasic extraction of 30 µL of plasma was achieved using 190/120/390 µL of MeOH/H_2_O/CHCl_3_. Samples have been centrifuged at 800× *g* for 20 min and the upper methanolic phase was collected, dried under Nitrogen flow and re-suspended in 100 µL of 20/80 (H_2_O/MeCN). The same internal standards mentioned above were added to the sample prior to extraction.

### 2.4. Liquid Chromatography–High-Resolution Mass Spectrometry

A 1200 infinity series high performance liquid chromatography (HPLC) system (Agilent Technologies, Santa Clara, CA, USA) coupled to an Exactive Orbitrap mass spectrometer (Thermo Fisher Scientific, Bremen, Germany) equipped with a heated electospray (H-ESI II) source was used. The HPLC separation was achieved using a SeQuant ZIC-HILIC 3.5 µm, 200 Å 100 × 2.1 mm (Supelco, Munich, Germany). The ESI source conditions and the MS tuning were the same of [22]. Full scan mass spectra were acquired from *m/z* 65 to 975 at a mass resolving power of 25,000 Full Width at Half Maximum (FWHM) at *m/z* 200. The MS instrument was set in dual polarity (positive/negative) acquisition mode.

### 2.5. Instrumental Calibration and Performance Control

Every second day instrument was stopped for cleaning, and recalibration was performed using a MSCAL6 ProteoMass LTQ/FT-Hybrid. Xcalibur 2.2 (Thermo Fisher Scientific, San Jose, CA, USA). Each intraday experiment has been performed after a general instrumental cleaning and recalibration.

The injection sequences were performed as follows: After column conditioning and equilibration with the injection of two analytical mobile phases, one extraction blank and 8 consecutive Quality Control samples (QCs), the samples were injected in random order, interspersed with one QC every 5 samples. Every 48 h of analysis the injection queue was stopped to clean the instrument, to recalibrate the instrument, clean the ion source and rinse the LC-MS system, to restore the initial instrumental performance. After re-conditioning the column through injections of QC samples, the sample sequences were restarted.

### 2.6. Chromatographic Columns (Figure 1)

Five different columns have been tested using the AF method prior the comparison with the AA method: (1) ACQUITY BEH-HILIC 1.7 µm, 2.1 × 100 mm (WATERS, Manchester, UK). (2) ACQUITY BEH-Amide, 1.7 µm, 2.1 × 100 mm (WATERS, Manchester, UK). (3) Halo Penta-HILIC, 2.7 µm, 2.1 × 100 mm (Advanced materials technology, Wilmington, DE, USA). (4) Luna HILIC 3 µm, 200 Å, 2 × 150 mm (Phenomenex, Le Pecq, France). (5) SeQuant ZIC-HILIC 3.5 µm, 200 Å, 2.1 × 100 mm, (Merck-Millipore, Fontenay sous-Bois, France).

### 2.7. Chromatographic Conditions

#### 2.7.1. Ammonium Acetate Method (AA)

In such method ([20]) the eluent B was 99:1 ACN:H_2_O with 10 mM of ammonium acetate, while eluent A was 100% H_2_O with 10 mM of ammonium acetate. The AA gradient started with 5% of A for 2.4 min then rising to 20% A at 5 min and to 40% A in 12 min to reach the 60% of A at 15 min, hold until the 18th minutes, going back to the initial condition at 19 min and equilibrating the column until the 27th min.

#### 2.7.2. Ammonium Fluoride Method (AF)

In such method the eluent B was 95:5 ACN:H_2_O with 2 mM of ammonium fluoride, while eluent A was 40:60 ACN:H2O with 2 mM of ammonium fluoride. The AF gradient started with 5% of A for 1.4 min then rising to 40% A at 3.5 min and to 70% A in 12 min to reach the 90% of A at 13 min, hold until the 16th min, and back to the initial condition at 17 min and equilibrating the column until the 24.5 min.

In both methods the column temperature was 35 °C, the flow rate was 0.4 mL/min, the injection volume was 10 µL.

### 2.8. Evaluation Process

The evaluation of the two chromatographic modifiers has been performed in three steps:

(1) Thirty-four analytical standards (Table 1) selected from different chemical classes typically present in the polar fraction of the human bio-fluids (amino-acids, phenolic acids, sugars, indoles purines and pyrimidines), exhibiting a logP in the range of −4 to 6, and a mass range from 112 to 449 Dalton, have been grouped in 6 clusters and injected using both chromatographic methods. Their respective performances have been evaluated by the scoring approach of Pezzatti et al. [23]. Briefly, this scoring approach allows to build a robust weighted score using the apparent retention factor, peak area, the signal to noise for each peak and a manual evaluation of the peak shape. Such scoring approach has been specifically developed for metabolomics experiments. The peak asymmetric factor and the peak resolution of some isomers has been also evaluated.

(2) To evaluate the matrix effect on the performances of the two chromatographic methods and highlight their intra-day variability, we performed for each method 16 consecutive injections of urine samples, evaluating the (i) identification, (ii) retention time shift, (iii) ionization efficiency (mean peak area) and (iv) relative standard deviation (RSD) of the peak area of the 34 analytical standards.

(3) Two complete sample-sets of already characterized [Narduzzi 2019] human bio-fluids (urine and plasma) have been analyzed using the ammonium fluoride method, each one consisting of about 5 days of consecutive analyses (about 210 injections in total). In order to assess the robustness of the developed protocol, several parameters have been taken into account to fit metabolomics requirements in terms of (i) system pressure at the beginning and the end of the runs, (ii) performance of the automatic integration of the internal standards (time shift and RSD of the peaks), (iii) statistical evaluation of the experiment to verify whether the different groups could be discriminated.

In the first two steps, peak integration has been performed semi-automatically using the Quan browser software in Xcalibur (Thermo Fisher Scientific, Bremen, Germany), while in the third step, data deconvolution has been automatically performed using xcms 3.2 [24]. Data sets were further pre-processed and normalized using NOREVA [25], then multivariate statistical analysis has been performed with Simca-P+ 13.0.2.

## 3. Results

### 3.1. Pre-Experimental Settings: Column Choice and Gradient Development

As electrostatic interaction is one of the leading process governing HILIC retention [4,26], the change of the additive salt might result in very different apparent retention factor of the stationary phase. Therefore, five different HILIC phases (BEH-AMIDE, BEH-HILIC, Penta-HILIC, Luna HILIC and the ZIC-HILIC) have been tested as preliminary work evaluating thus the range of detected analytical standards (*n* = 34) in the corresponding chromatograms.

The BEH-AMIDE and ZIC-HILIC columns performed similarly; considering the number of retained and identified compounds, the BEH-AMIDE performed better in the separation of the sugars, while the ZIC-HILIC had a better performance for the acids. The BEH-AMIDE showed very narrow peaks with a low point-per-peak number due to the low scanning rate of our MS instrument (Appendix A). The elution speed of the BEH-AMIDE column was considered not compatible with our Exactive acquisition frequency and therefore it was discarded. The Penta-HILIC showed a lower performance than the former columns, with a fast elution time (Appendix A) but also a bad selectivity regarding acidic compounds (Appendix A). The Luna HILIC and the BEH-HILIC showed the worst performance with no retention for several compounds. The ZIC-HILIC column was therefore selected as it showed an average best retention for all classes of metabolites, with a chromatographic resolution compatible with our MS instrument.

In this study, a neutral pH has been implemented for the AF chromatographic method, because the pKa for ammonia (9.5) was out of the working range of the ZIC-HILIC column (3–8), while the acidification of ammonium fluoride solutions might be hazardous for the operators’ health. The gradient of the chromatographic run for the ammonium fluoride method was derived from the ammonium acetate method, slightly modifying the total run time (from 27.0 to 24.5 min, Figure 1), because we observed a general earlier elution of the compounds, combined with better separations (Figure 2).

### 3.2. Method Comparison: Standard Injection Evaluation

A total of 34 different analytical standards (Table 1) representing different chemical classes have been analyzed using both chromatographic methods. Respective chromatographic performances have been assessed applying an approach specifically designed to evaluate metabolomics chromatographic methods. It includes the column apparent retention factor, peak shape, peak intensity and the signal to noise ratio under a single score (S_i_ score [23]), making it convenient to compare different methods, not matter the stationary phase involved; the final score is a value ranging from 0 (no peak detected) to 1 (best peak characteristics). The results of this analysis are reported in the Appendix A. A simplified version is reported in Table 2.

The mean asymmetric factor was slightly better for the AF method, 1.73 vs. 1.96 for the AF and AA methods respectively (Appendix A). The resolution of isomeric peaks showed to be similar between the two methods, with contrasting results depending on the class of the compounds: the amino-acids leucine and isoleucine were better separated by the AA method, while sugars (sorbitol and mannitol) were better separated by the AF method (Appendix A). The retention factor was observed as slightly lower in the AF method, also because the chromatographic method was shorter.

In positive ionization mode, as shown in Table 2, a major improvement in compound detectability was obtained applying AF conditions vs. AA ones. Three compounds were only detected in the AF chromatographic run: uracil, 2-hydroxy-indole acetic acid and 5-hydroxy-indole acetic acid. Further, all the detected compounds showed better S_i_ score under AF conditions, with an average improvement of 0.24. The sharp difference between the methods’ performances was mostly due to parameters related to peak height, peak area and signal to noise ratio which were significantly improved under AF conditions; as an illustration, the median value of peak area of the AF method is 8.1 times higher than the median value of the AA method (Table 3).

In negative ionization, the results were less univocal than using the positive ionization. Overall the AF method showed an average improvement of 0.19 compared to the AA method, but analyzing the data in detail, some contrasting results were obtained. While 18 compounds showed better characteristics in the AF conditions, with a sharp improvement in almost all cases (mean improvement 0.44), better results in the AA conditions have been observed for 11 further compounds, with a mean decrease of 0.22; three of those exhibited a minimal change ≤ 0.06 (l-phenyl-lactic acid, myo-inositol and fumaric acid), while 4 presented a sharp improvement ≥ 0.34 (uracil, taurine, 3-indole acetic acid and hippuric acid). The differences observed in the negative ionization is mostly due to the peak height and peak area; in this case the median value of the peak area of the AF method was 3.2 times higher than the median value of the AA standards (Table 3).

From the ionization efficiency’s point of view, it is important to compare the ionization intensity between positive and negative modes within the AA and AF conditions. In the AA method, the negative ionization of the compounds is in median 2 times higher than in positive ionization (Table 3); this could be one of the main reason why generally negative ionization is considered most effective in metabolomics than the positive one. On the other hand, the AF method equilibrates this situation, reporting a higher median peak area of the common compounds for positive ionization (Table 3). To summarize, the AF method shows to increase the ionization in comparison to the AA, and it equilibrates the ionization efficiency of the ESI+ and ESI– (Table 3).

### 3.3. Repeatability and Robustness Test: Pooled Sample Injections Evaluation.

Aliquots from all the 155 urine samples obtained from the experiment of Narduzzi et al. [27] were mixed to constitute a representative urine pooled sample to perform method evaluation using repetitive injections. After instrumental conditioning, obtained through the injection of three analytical mobile phases, the pooled samples have been injected 16 times in each chromatographic method. As most of the 34 standards injected in the previous experiment are common endogenous metabolites present in urine, a targeted search for their signals aiming to evaluate the methods repeatability in terms of RT shift and peak area was performed. Analytes’ ionization efficiency was also assessed to check for potential matrix effects on the ionization ability. Please note that we did not spike pure compounds in the pooled urine sample, to avoid to reach unrealistic compound concentrations in urine.

The results of this trial are reported in Table 3. A similar RT shift has been observed for both methods, indicating that the AF does not influence the intraday repeatability of the retention times. It could be observed that the AF conditions allowed retrieving a higher proportion of compounds in the positive ionization mode compared to AA conditions, probably due to a lower ion suppression effect (Figure 3). On the other hand, selecting only the compounds detected in both chromatographic methods (*n* = 23), the median intensity was higher in the AF method in both ESI+ (1.4 times) and ESI– (about 4.5 times) ionization modes. The AF method also performed better regarding repeatability of the measurements: The fairly median RSD of the peaks’ area was lower in both positive and negative ionization modes (Table 4). This could be explained by the higher ionization efficiency allowed using AF; indeed, a higher MS signal has higher probability to fall within the linear range of the instrument, giving thus a more reliable estimation of the real concentration of the analytes.

### 3.4. Evaluation on Real Datasets: Urine and Plasma Metabolomics

The AF method scored best in the previous tests; therefore, we decided to perform two real metabolomics experiments using this method, to evaluate its inter-day repeatability and its consistency with automatic deconvolution software (*xcms*). The sample-sets, already characterized using a classical RPLC-metabolomics workflow [27], consisted in urine and plasma samples collected from volunteers doped with micro-doses of different hormones mixtures. Each sample-set consisted of 155 samples. Considering the necessary blanks and QC injections, the analysis of each sample-set consisted of about 210 consecutive injections; within the range of common metabolomics experiments.

First, back-pressure at the beginning and the end of the runs has been taken in account to evaluate AF method applicability. Two different ZIC-HILIC columns have been used is this study, one for each sample-set. The first sample-set (urine) was analyzed using the same column on which the previous tests have been performed. This ZIC-HILIC column exhibited a backpressure range of 120–220 bars (gradient-wise) at the beginning and 140–225 bars at the end of the injection sequence. The second ZIC-HILIC column has been used for the analysis of the plasma samples; it showed a lower backpressure range at the beginning (85–160 bars), and an augmented range 145–225 after conditioning. The shift in column back-pressure did not affect the column apparent retention factor. The mean RT shift observed for the internal standards remained consistent between the two columns, with 14 s for the first column and 16 s for the second one. This shift only represents about 1% of the total time, a similar value that is commonly accepted for RP chromatography [28]. When switching from the old to new column the retention time shift was in average of 8.5 s (Appendix A), within the acceptable range.

Automatic data deconvolution was achieved through *xcms* software. In theory, if any RT or intensity shifts would occur in the data set or if the peaks shape would be modified, the automatic processing should not be able to correctly integrate the signals, or to correctly group the peaks during the retention time correction process, consequently leading to high peak area RSD. Such issue was not observed in the present study; *xcms,* indeed, correctly integrated the internal standards signals in both sample-sets and in both ionization modes. Corresponding mean peak area RSDs remained below 20% (19.9%, 16.6%, 19.6% and 18.85% for urine pos-neg and plasma pos-neg characterization, respectively), indicating a satisfactory stability of the analytical system, providing expected inter-day precision (about 5 days of analysis) within such metabolomics workflow.

The statistical analysis of the datasets was performed using PLS-DA [29]. As shown in Figure 4, the PLS-DA could distinguish both treatments in the two investigated bio-fluids, except for the positive ionization of urine samples (data not shown). Especially in plasma, the separation of the groups was sharp (*p*-value < 2 × 10^−14^ and 3 × 10^−29^ in ESI+ and ESI–, respectively), indicating a relevant influence of the treatment on the subjects’ metabolism. While similar performances in terms of groups separation had also been observed in the previous study relying on RP chromatographic separation for the plasma samples, the HILIC-based protocol allowed discriminating urine sample groups, which was not the case using RP separation. Such results relying on AF enhanced HILIC chromatography highlight both applicability and relevance of the proposed metabolomics workflow, which furthermore, appears as complementary to classical RP method allowing extended metabolome coverage for deeper and subsequent pathway investigation.

### 3.5. Tip and Tricks: Method Adjustments and Important Details

Developing a novel chromatographic method, requires taking into account several critical parameters, which are finally not often reported, detailed and discussed in scientific communications. Nonetheless, those practical details are important because they strongly influence the final performance and applicability of a method. For this reason, the following section provides a list of key points to be carefully considered when working with an ammonium fluoride method in HILIC chromatography.

Needle and injector wash: Although the washing method is often overlooked during the method development [30], it was observed to be a critical parameter within the present study. Ammonium Fluoride solubility in organic solvent is very low, while it is very soluble in water. Accumulation in the tubing may become problematic after 10 to 20 injections without a proper rinsing program, determining problems in the tubing, especially in the high organic phase one. We suggest to rinse the needle and the syringe with high amount of water between every injection (at least 3 syringe volumes each).

Column conditioning: Switching from a salt to another one may be problematic. In this experiment we have used two columns; the first and older one had already been used in previous experiments with ammonium acetate as additive (so the conditioning was performed with AA). The second and new one was first conditioned and used with ammonium fluoride. While trying to switch back to AA method using the second one, very broad peak shapes for cationic compounds, like amines, were observed. Such effect was supposed to be due to the fact that the acetate group of the AA cannot compete with the electronegativity of the fluorine group bound to the betaine group of the stationary phase, limiting the eluting strength of the acetate group. A similar modified retention factor has been reported by Pesek et al. [18] when switching modifiers in aqueous normal phase chromatography. We suggest to be consistent with the choice of the column and the associated modifier.

Neutral vs. acidic conditions: Salts like ammonium acetate and ammonium fluoride exhibit a pH of 6.8 and 6.6 respectively in water. Nonetheless, their acidic pKa is far from these points and they consequently not serve as buffering agents (4.7 and 3.2 respectively) [5]. Some might be tempted to add hydrofluoric acid or any other acidic solution to reach the pKa and obtain thus a buffering solution and facilitate the protonation effect to enhance ESI+ ionization. Nevertheless, the fluorine ion obtained from the pH adjustment is rather dangerous and toxic. It might corrode tubing or the ZIC-HILIC column itself. Therefore, no pH adjustment should be allowed. Neutral pH is recommended.

## 4. Conclusions

HILIC chromatography is a very interesting alternative analytical strategy to widen the metabolome coverage in LCMS based metabolomics workflows. In metabolomics, ammonium formate/acetate are the most common modifiers for HILIC chromatography, but they are known to have some limitations, with several compounds known to be poorly ionized under such conditions.

As ammonium fluoride is reported to be a better ionization agent in RP and HILIC chromatography [14,18], in this study we tested AF as additive salt within a HILIC-based metabolomics workflow, developing an ad-hoc AF method. We evaluated its overall performances in comparison to a robust ammonium acetate method through two different tests. Then, we tested the AF method robustness and consistency performing two real metabolomics experiments. The AF method showed to be superior to a standard AA method and overall satisfying in terms of reproducibility and robustness. Ionization of several compounds improved sharply, repeatability was improved, peak asymmetry improved slightly, while the retention factor was only slightly affected. The method showed a good repeatability in both intra-day and inter-day tests, and when subjected to automatic data processing, no deleterious effects could be noted in the process outcome.

## Figures and Tables

**Figure 1 metabolites-09-00292-f001:**
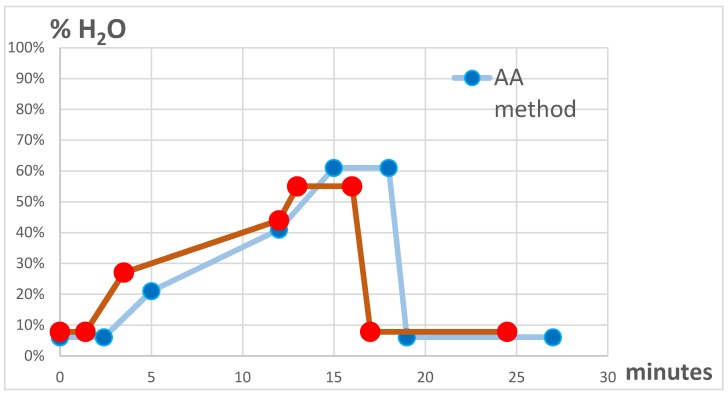
Chromatographic gradients used in the Ammonium Acetate (AA) and the Ammonium Fluoride (AF) methods.

**Figure 2 metabolites-09-00292-f002:**
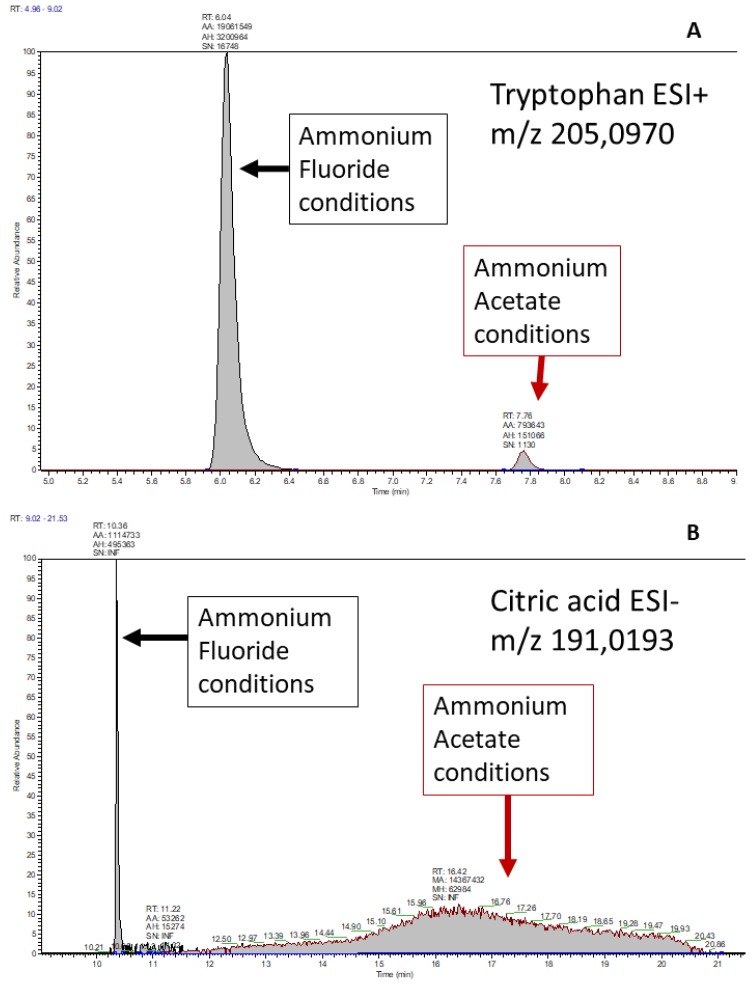
(**A**,**B**) two examples of different peak shape between the two chromatographic methods, reporting tryptophan (up) and citric acid (below). In the first case, tryptophan showed higher intensity and peak area, but similar peak shape and lower apparent retention factor in the AF method. In the second case, in the AA method the signal of the citric acid is present, but no automatic detection software is able to integrate correctly such 10 minute-wide peak. In contrast, the AF method shows a fairly nice peak for citric acid.

**Figure 3 metabolites-09-00292-f003:**
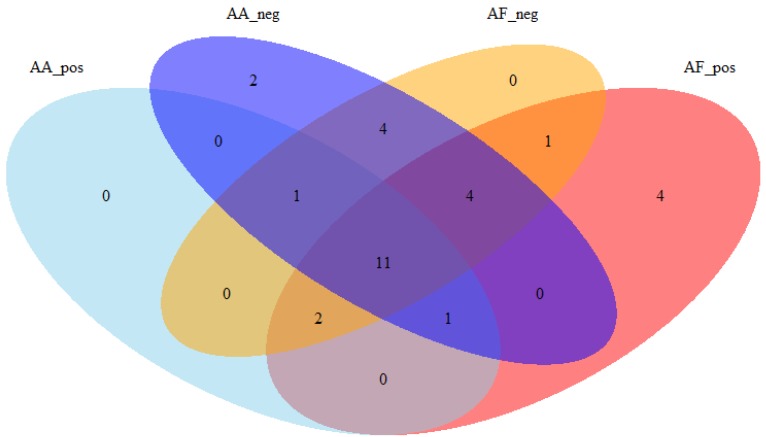
Venn diagram reporting the number of common compounds detected across the various methods.

**Figure 4 metabolites-09-00292-f004:**
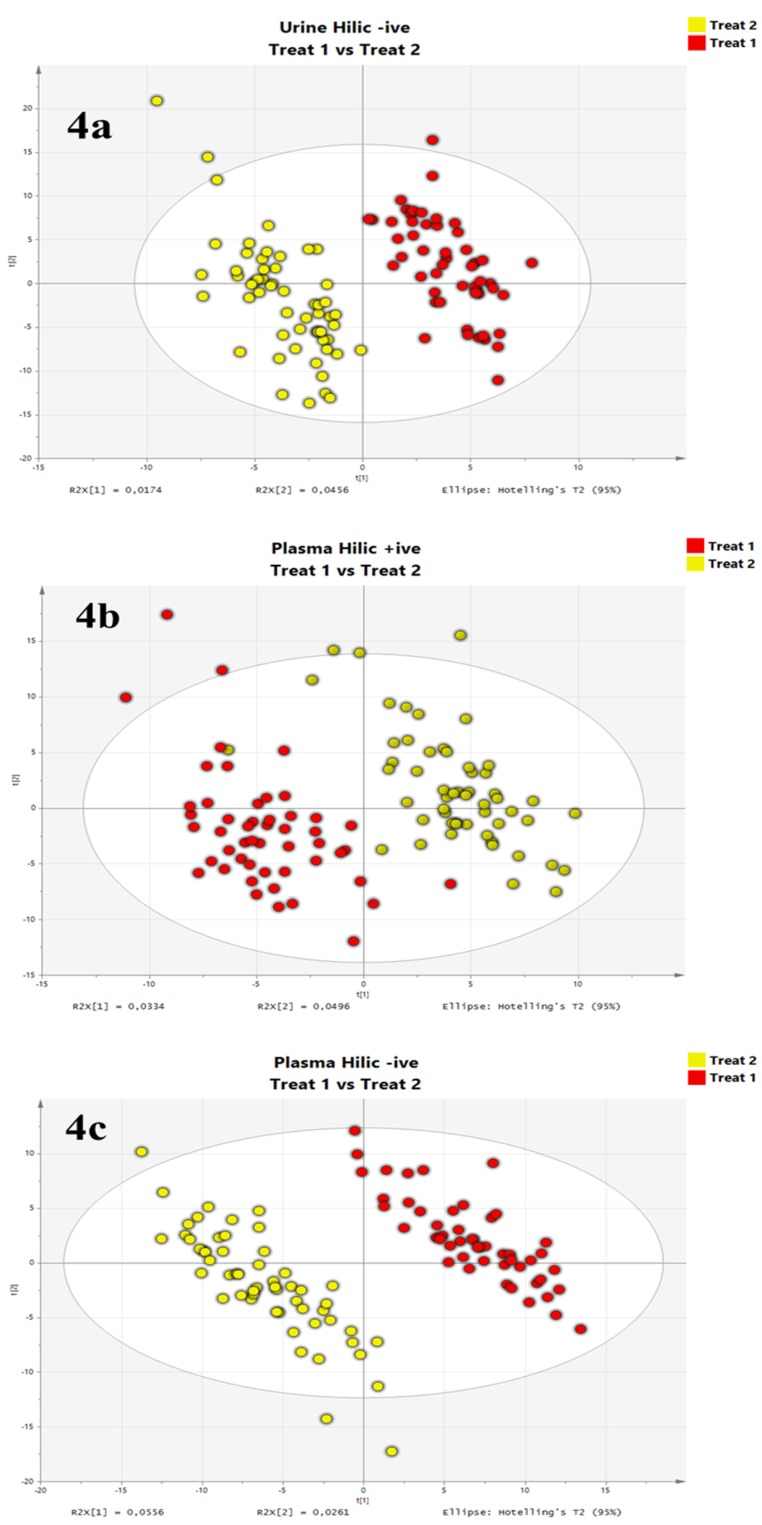
Scatter plots of the PLS-DA performed on (**A**) urine (ESI–) and (**B**,**C**) plasma (ESI+ and ESI–, respectively) datasets after analysis using the HILIC-AF method.

**Table 1 metabolites-09-00292-t001:** List of 34 analytical standards used in this study. The list includes compounds of different classes, the monoisotopic mass ranging from 112 to 449 Dalton and a logP ranging from −4.7 to 6.

Compounds	logP	Mass Da	Class
sorbitol	−4.7	182.0790	sugar
mannitol	−4.7	182.0790	sugar
betaine	−3.3	117.0790	amino acids
nicotinamide n-oxide	−2.7	138.0430	pirimidine
taurine	−2.5	125.0147	amino acids
myo-inositol	−2.1	180.0634	sugar
inosine	−1.9	268.0808	purine
cis-4-hydroxy-d-proline	−1.8	131.0583	amino acids
d-fructose-6-phosphate	−1.8	260.0297	sugar
citric acid	−1.7	192.0270	acid
uridine	−1.7	244.0695	pirimidine
2’-deoxyguanosine-5-phosphate	−1.4	345.0485	purine
d,l-pantotheic acid	−0.9	219.1107	vitamin
glu-val-phe	−0.9	393.1900	amino acids
uracil	−0.7	112.0273	pirimidine
hydroxy-hippuric acid	0.0	195.0532	phenolic acid
fumaric acid	0.0	116.0110	acid
adipic acid	0.1	146.0579	lipid
biotin	0.1	244.0882	vitamin
valine	0.2	117.0790	amino acids
5-hydroxy-indole acetic acid	0.3	190.0504	indole
2-hydroxy-indole acetic acid	0.3	190.0504	indole
hippuric acid	0.3	179.0582	phenolic acid
leucine	0.7	131.0947	amino acids
isoleucine	0.7	131.0947	amino acids
l-tryptophan	1.0	204.0880	amino acids
l-3-phenyllactic acid	1.1	166.0630	phenolic acid
l-kynurenine	1.1	208.0848	phenolic acid
l-phenylalanine	1.1	165.0791	amino acids
3-indole acetic acid	1.4	175.0633	indole
d-homophenylalanine	1.5	179.0946	amino acids
oxooctanoyl homoserine lactone	2.0	297.1940	amino acids
glycourso-deoxycholic acid	3.5	449.3135	bile acid
tetradecanoic acid	6.1	228.2089	lipid

**Table 2 metabolites-09-00292-t002:** Comparison of methods performances on the basis of Si score [19] for the set of *n* = 34 analytical standards. The delta score between the two methods is calculated as the AF score – the AA score. The blue background color indicates AF score >> AA score (> 0.3). The green background color indicates the AF score > AA score. The red background color indicates AF score < AA score.

Compounds	Pos. Ionization S_i_ Score			Neg. Ionization S_i_ Score		
	AA	AF	Δ Score	AA	AF	Δ Score
betaine	0.06	0.58	0.52	0.00	0.00	0.00
valine	0.04	0.35	0.32	0.13	0.57	0.44
leucine	0.05	0.39	0.34	0.10	0.36	0.26
isoleucine	0.07	0.36	0.29	0.10	0.43	0.33
l-phenyllactic acid	0.00	0.00	0.00	0.14	0.10	–0.04
myo-inositol	0.00	0.00	0.00	0.65	0.64	–0.01
fumaric acid	0.00	0.00	0.00	0.42	0.36	–0.06
fructose-6-phosphate	0.23	0.43	0.20	0.39	0.23	–0.16
citric acid	0.00	0.00	0.00	0.00	0.28	0.28
uracil	0.00	0.07	0.07	0.38	0.04	–0.34
nicotinamide-n-oxide	0.12	0.20	0.09	0.55	0.39	–0.16
uridine	0.06	0.20	0.14	0.33	0.43	0.11
5-hydroxy-indole acetic acid	0.00	0.29	0.29	0.43	0.90	0.47
taurine	0.17	0.35	0.18	0.72	0.36	–0.35
mannitol	0.00	0.00	0.00	0.00	0.88	0.88
sorbitol	0.00	0.07	0.07	0.09	0.88	0.80
hydroxy-hippuric acid	0.00	0.00	0.00	0.05	0.20	0.15
tetradecanoic acid	0.00	0.00	0.00	0.00	0.10	0.10
3-indole acetic acid	0.00	0.00	0.00	0.53	0.06	–0.46
hippuric acid	0.01	0.14	0.14	0.67	0.30	–0.37
glycourso-dexoycholic acid	0.07	0.35	0.28	0.28	0.83	0.55
inosine	0.09	0.65	0.56	0.14	0.73	0.58
d-homophenylalanine	0.18	0.73	0.56	0.21	0.74	0.54
l-phenylalanine	0.15	0.70	0.55	0.20	0.74	0.54
l-kynurenine	0.16	0.71	0.55	0.16	0.63	0.47
biotin	0.14	0.66	0.52	0.14	0.65	0.51
d-,l-pantotheic acid	0.00	0.00	0.00	0.29	0.72	0.43
glu-val-phe	0.16	0.38	0.22	0.23	0.70	0.47
l-tryptophan	0.17	0.69	0.52	0.00	0.74	0.74
d,l-pantotheic acid	0.11	0.62	0.51	0.00	0.00	0.00
adipic acid	0.00	0.00	0.00	0.38	0.10	–0.28
cis-4-hydroxy-d-proline	0.16	0.75	0.59	0.57	0.37	–0.19
2-hydroxy-indole acetic acid	0.00	0.67	0.67	0.00	0.00	0.00
deoxyguanosine-5-phosphate	0.00	0.00	0.00	0.44	0.58	0.14

**Table 3 metabolites-09-00292-t003:** A comparison of the median area of the different ionization modes in the two methods.

Chromatographic Conditions	AA		AF	
Ionization Mode	ESI+	ESI–	ESI+	ESI–
median area	5,234,608	10,394,744	42,179,020	35,910,970
ratio neg/pos	2.0		0.9	
ratio AFpos/AApos	-	-	8.1	-
ratio AFneg/AAneg	-	-	-	3.5

**Table 4 metabolites-09-00292-t004:** Results of the repeatability test.

Ionization Mode	*ESI+*		*ESI–*	
Additive	AA	AF	AA	AF
ND** peaks	14	7	11	11
Median peak area	692,901	1,036,833	663,922	2,994,235
Median area RSD*	29 max–min 49–8	10 max–min 53–2.8	17 max–min 90–6	8.7 max–min 45–5
Mean RT shift	11 s	10 s	16 s	9 s

*Relative Standard Deviation. **Not Detected.

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
