# Peer review of "Ammonium Fluoride as Suitable Additive for HILIC-Based LC-HRMS Metabolomics"

_metabolites, 2019, doi:10.3390/metabo9120292_

Round 1
Reviewer 1 Report
Overall the submission provides a considerable amount of information on the use of HILIC for metabolites using ammonium fluoride as an additive. However, there are some issues that need to be addressed by the authors:
Reference 18 cited in the Introduction is not HILIC but aqueous normal phase (ANP). The two separation modes are not the same since HILIC is based most often on a partition mechanism and ANP is based on an adsorption/desorption mechanism. Adequate document of the ANP mechanism can be found in the following reference: J. Phys. Chem Part B, 119 (2015) 3063-3069. Therefore, the authors should correct this error so that readers do not assume that ANP and HILIC are the same.
The most common designation by people in the field of chromatography is for the A solvent to be water and the B solvent to B organic. It would be best to follow this accepted format.
Why did the results reported here show varying results with respect to AA vs. AF when the data reported in Ref. 18 (ANP) showed only improvement when using AF?
Why was it necessary to clean the instrument every second day? Recalibration frequently seems reasonable but instrument cleaning does not. What was the cleaning done? Ion source?
Was the high RSD for area measurements because no internal standard was used?
What about interday repeatability? Also, is there a trend from day to day either to shorter or longer retention times?
What about column to column repeatability for the same stationary phase? Can one buy the same stationary phase and obtain very similar results from previous measurements?
Reviewer 2 Report
In their manuscript the authors describe the advantages of using ammonium fluoride as additive for HILIC based LC-HRMS.
In general the authors tackle a timely topic where HILIC is an essential chromatographic tool in metabolomics, but results often in bad separation and intra/inter batch variations. Using ammonium fluoride seems to improve the chromatography substantially, especially in positive mode.
Remarks:
despite the fact that the authors describe the improvement in positive mode, I wondered whether they checked negative mode more extensively. From a metabolomics point of view, a lot of metabolites are best measured in negative ion mode (for instance phosphates, organic acids etc). Did the authors check whether increasing the pH of the ammonium fluoride buffer to pH 9-9.5 using for instance NH4.OH resulted in a significant improvement? I believe that the section of the pH should be placed more upfront, as it is now positioned at the end of the manuscript, while I believe it is a very essential section of the manuscript. I would like to see the improvement of the ammoniumfluoride on actual cellular extracts, looking at for instance ATP, ADP and AMP, ... also NAD/NADH and NADP/NADPH are of critical interest to many users. does the ammoniumfluoride affect any biological ratio's such as glutathione oxidation state (GSSG/(GSH+GSSG)) or the energy charge ((ATP+0.5*ADP)/(ATP+ADP+AMP)), ...? It would be essential to check these ratio's in the AA versus AF setups. as HILIC is used for lipids as well, is there any improvement here as well? does the detectibility of positively (and negatively) charged lipids increase using ammonium fluoride, in terms of ionisation and separation?Minor
1. the text mentions ammonium formiate a couple of times, do the authors mean ammonium formate?
2. text contains some other typo's such as LIquid in stead of Liquid ...
3. how abrasive is the fluoride for the MS? Do the vendors mention anything about its usage?
Round 2
Reviewer 1 Report
None at this time.